# HCV RNA Quantification by a Domestic Commercial Assay: A Case Study among People Who Inject Drugs in Vietnam

**DOI:** 10.3390/diagnostics13223456

**Published:** 2023-11-16

**Authors:** Quynh Bach Thi Nhu, Linh Le Thi Thuy, Hong Thi Nguyen, Binh Nguyen Thanh, Delphine Rapoud, Catherine Quillet, Hong Thi Tran, Roselyne Vallo, Thanh Nham Thi Tuyet, Laurent Michel, Laurence Weiss, Philippe Vande Perre, Vinh Vu Hai, Nicolas Nagot, Oanh Khuat Thi Hai, Don Des Jarlais, Huong Thi Duong, Khue Pham Minh, Didier Laureillard, Jean-Pierre Molès

**Affiliations:** 1Public Health Faculty, Hai Phong University of Medicine and Pharmacy, Hai Phong 180000, Vietnam; btnquynh@hpmu.edu.vn (Q.B.T.N.); lethuylinh1189@gmail.com (L.L.T.T.); nguyenhongshpt@gmail.com (H.T.N.); nguyenducbinh.yhp@gmail.com (B.N.T.); tranthihong42@gmail.com (H.T.T.); dthuong@hpmu.edu.vn (H.T.D.); pmkhue@hpmu.edu.vn (K.P.M.); 2Pathogenesis and Control of Chronic and Emerging Infections, University of Montpellier, Inserm, 34394 Montpellier, France; delphine.rapoud@yahoo.fr (D.R.); catherine_quillet@yahoo.fr (C.Q.); roselyne.vallo@umontpellier.fr (R.V.); p-van_de_perre@chu-montpellier.fr (P.V.P.); n-nagot@chu-montpellier.fr (N.N.); didier.laureillard@yahoo.fr (D.L.); 3Supporting Community Development Initiatives, Hanoi 111000, Vietnam; thanhnham@scdi.org.vn (T.N.T.T.); oanhkhuat@scdi.org.vn (O.K.T.H.); 4Inserm UMRS 1018, Pierre Nicole Center, French Red Cross, 75005 Paris, France; laurent.michel@croix-rouge.fr; 5Université Paris Descartes, Sorbonne Paris Cité, Inserm U976, 75006 Paris, France; laurence.weiss@aphp.fr; 6Infectious and Tropical Diseases Department, Viet Tiep Hospital, Hai Phong 180000, Vietnam; vinhvuhai@gmail.com; 7School of Global Public Health, New York University, New York, NY 10012, USA; don.desjarlais@nyu.edu; 8Infectious Diseases Department, Caremeau University Hospital, 30900 Nîmes, France

**Keywords:** diagnostic, HCV, drug users, Vietnam, assays

## Abstract

The desired performance of nucleic acid testing (NAT) may vary if used for disease diagnosis or for the evaluation of the therapeutic efficacy of a treatment, although in most cases, the same assay is used. However, these tests may not be affordable in many situations including in low/middle income countries that in response have developed domestic assays. Given the example of HCV NAT among people who inject drugs in Vietnam, we aimed at evaluating a domestic assay versus an FDA- and CE-approved assay. This cross-evaluation revealed that (i) the domestic assay had a poorer sensitivity with a threshold of detection above 10^4^ IU/mL, and (ii) the FDA-approved assay had a percentage of false negative results close to 1%. Together, in the present study, the domestic assay had a performance compatible with diagnosis purposes (given that this population was 70% HCV seropositive) but not compatible with HCV treatment monitoring (given that treatment failures are rare and the observed viremia frequently below the threshold of detection). This study highlights the need for a proper evaluation of HCV RNA domestic assays in order to efficiently contribute to the WHO HCV elimination target by 2030.

## 1. Introduction

An estimated 71 million people have chronic HCV infection worldwide, with a condition severity ranging from asymptomatic carriage to a serious, lifelong illness. In 2016, it was estimated that 400,000 people died from hepatitis C, mostly from cirrhosis and hepatocellular carcinoma [1]. The prevalence varied by region but could be as high as 4.7% of the adult population in Gabon [2]. The epidemic is still on-going, mainly driven by occupational or unsafe injection, with a global annual incidence of 23.7 per 100,000 population (95%CI: 21.3–28.7) [1]. Since 2014, new antiviral treatments, namely Direct-Acting Antivirals (DAA) can cure more than 95% of persons with HCV infection, increasing the possibility for elimination of hepatitis C. In 2015, the WHO adopted a global hepatitis strategy to eliminate viral hepatitis by 2030 (90% reduction in HCV incidence, 65% reduction in mortality, 80% of individuals with positive HCV results treated) [3]. 

The current diagnostic of chronic HCV infection relies on plasma RNA detection by nucleic acid testing (NAT). The treatment success is also evaluated by quantifying the plasma HCV RNA viral load. Current WHO recommendations state that such tests should have a sensitivity and a specificity above 98% and a threshold of detection above 1000 copies per mL or 400 IU/mL [4]. Most central laboratory platforms such as ABBOTT RealTime^®^ or ROCHE COBAS^®^ Taqman^®^ provide tests with these diagnostic performances. Also, near point-of-care techniques such as the GeneXpert^®^ system are available for use outside of a laboratory environment. In low-middle incomes countries (LMIC), however, HCV RNA quantification assays are not commonly used, performed in centralized laboratory hence distant from the patient and are frequently unaffordable. Consequently, in order to improve accessibility, LMIC have frequently developed their own domestic assays on open platforms. These assays do not always satisfy international qualification procedures and use different sets of reference panels. Nevertheless, they benefit from a tailored design to the locally circulating viral strains, which are sometimes under-represented in the reference panels. Furthermore, they benefit from an excellent knowledge of the population to test, which may differ from the standard population in term of contaminants, i.e., inhibitors of NAT assays such as substances interacting directly with DNA or blocking the activity of the polymerase and reverse transcriptase or other PCR mixture components (e.g., MgCl_2_) [5]. 

In Vietnam, almost 1 million people live with chronic HCV infection, making Vietnam the 12th most affected country in the world [6]. The genotype (GT) 6 is overrepresented, followed by GT1a and GT1b [7]. In the absence of widely available DAA treatment in the country, it is estimated that over 70% of antibody seropositive people are viremic [6]. Certain key populations are more likely to be infected, among them people who inject drugs (PWID). Recent surveys carried out in Hai Phong city, a city inhabited by 2 million with an estimated PWID population of 5000 [8], showed a prevalence in HCV positive serology of 70% [9,10]. As the latter are often poly-drug users, NAT may not perform as well as in the general population because of the presence of NAT inhibitors (psychoactive drugs, polyphenols (tea), polysaccharides (algae, bivalves), antiretroviral drugs, … or any combination thereof) [11]. In most hospitals across the country, the diagnostic of chronic HCV infection was performed with the Light Power iVA HCV qPCR assay (VietA Corp., Hanoi, Vietnam), a domestic NAT assay combining a random reverse transcription and a specific real-time PCR. According to the manufacturer, this assay has the following performances: sensitivity 100%, specificity 100% and a lower limit of quantification of 300 cp/mL (or 120 IU/mL). 

Our study aims to compare the performance of an international FDA- and CE-approved technique, namely GeneXpert^®^ assay, with the domestic Light Power iVA HCV qPCR assay.

## 2. Materials and Methods

### 2.1. Study Design

We implemented a diagnostic accuracy study. All samples underwent the studied test, Light Power iVA HCV qPCR assay, and the reference standard, the GeneXpert assay. 

### 2.2. Study Population and Study Conduct

Participants were enrolled between October 2018 and January 2019 in two large studies (DRIVE and DRIVE-C, NCT03526939 and NCT NCT03537196) aiming to control HIV and HCV epidemics among PWID in Haiphong, Vietnam, by combining mass screening through successive respondent-driven sampling surveys and linkage to care [7,8]. These studies were carried out in the offices of community organizations (CO), i.e., outside the healthcare system, because the target population is reluctant to go to hospital. In addition, all the activities of these studies were carried out by community workers (mostly former injecting drug users) and commissioned nurses from the provincial HIV reference laboratory. Eligible participants had to be 18 years of age or older, be an active drug injector (skin injection marks and positive urine test for heroin), have a positive HCV serology and be willing and able to provide informed consent for these studies. The first 600 participants of the third mass screening using respondent-driven sampling method were included in the present study.

### 2.3. Sample Collection and Processing

For each participant, urine samples were collected at the study sites and immediately tested for drug detection using Drug-screen Multi 7A carte (Nal von Minden, Regensburg, Germany). The assay detects heroin/morphine, methamphetamine, methadone, cannabis, benzodiazepine, ketamine and ecstasy.

For each participant, 9 mL of venous whole blood was collected in EDTA tubes. After centrifugation, three aliquots of plasma were prepared. Two tubes were stored in a sample repository at −80 °C, one tube was transported within 24 h to the high-tech center laboratory, Hai Phong University of Medicine and Pharmacy and used for HCV VL testing. Samples were transported daily in a cooler box between 2 and 10 °C (a half-hour transport).

### 2.4. HIV and Hepatitis Biological Assessments

On-site blood testing was conducted by study nurses. HIV serology diagnostic followed national guidelines including an initial rapid test using SD BIOLINE HIV 1/2 3.0 (Standard Diagnostic Inc., Suwon-si, Republic of Korea), followed by two other rapid tests (Alere Determine HIV 1/2 (Abbott Alere Medical Co., Waltham, MA, USA) and VIKIA^®^HIV1/2 (Biomerieux, Lyon, France) for confirmation. HCV serology relied on a rapid test SD BIOLINE HCV (SD Standard Diagnostic Inc., Suwon-si, Republic of Korea). CD4 counts were performed using the BD FACS Count system (BD Biosciences, San José, CA, USA) and HBV active infection was assessed through HBsAg detection using VIKIA^®^ HBsAg ELISA (Biomérieux, Lyon, France) at the provincial HIV reference laboratory.

### 2.5. HCV RNA Quantification by Light Power iVA HCV qPCR Assay

Upon reception, total RNA was extracted from 200 µL of plasma by Invisorb Spin Universal Kit (Stratec Molecular, Berlin, Germany). The RNA extracts were tested for quality control using Nanodrop technology (Thermo Fischer Scientific, Waltham, MA, USA). All samples had an A260/A280 ratio between 1.8–2.0 and RNA concentration between 18 and 25 ng/µL.

RNA was synthesized into cDNA by reverse transcriptase enzyme (iVA cDNA synthesis kit, VietA Corp., Hanoi, Vietnam). Briefly, 15 µL of total RNA extracts was mixed with 8 µL RT-Mix, then incubated according to the following program: 25 °C for five minutes, 42 °C for 30 min, 85 °C for five minutes. A total of 2.5 µL of cDNA products was used to perform real-time PCR reaction based on specific primers targeting the 5′UTR region (Light Power iVA HCV qPCR, VietA Corp., Hanoi, Vietnam). The sequence of the primers was not provided by the manufacturer. Each PCR plate consisted of sample in monoplicate and four standards of HCV sequence ranging from 40–40,000,000 IU/mL (10^2^ to 10^8^ copies/mL declared by the manufacturer).

### 2.6. HCV RNA Quantification by Xpert^®^ HCV VL

The procedure for performing Xpert^®^ HCV VL (Cepheid, Sunnyvale, CA, USA) was implemented according to the manufacturer’s instructions. In brief, one mL of plasma was transferred into the sample chamber of the cartridge, which contained all the reagents to perform RNA extraction, RT reaction and real-time PCR amplification. The detection range was from 10 to 100,000,000 IU/mL.

### 2.7. HCV Genotyping

The HCV genotype was assayed for all participants with an HCV VL > 1000 cp/mL. Briefly, from the same RNA extraction product as for HCV RNA quantification, a one-step RT PCR (Qiagen, Les Ulis, France) using NS-5B gene-specific primers (5′-TAT GAY ACC CGC TGY TTT GAC TC and 5′-GCN GAR TAY CTV GTC ATA GCC TC) at 600 nM was carried out. A nested PCR was performed on 5 µL of the RT-PCR reaction using the following primers (5′-CTG YTT TGA CTC MAC NGT MAC and 5′-CAT AGC CTC CGT GAA GRC TC) at 300 nM and the Advantage-2 Polymerase (Ozyme, Saint-Cyr-l’École, France). After amplicon purification, sequences were obtained using BigDye terminator V3.1 cycle sequencing kit (Thermo Fisher Scientific, Waltham, MA, USA) and analyzed on an ABI PRISM 377 DNA Sequencer [12]. HCV genotype were determined using geno2pheno (HCV) v. 0.92 [13].

### 2.8. Data Collection and Data Analyses

Data were collected using a face-to-face questionnaire administered by trained community workers on socio-demographic characteristics, injecting and drug use. Data were collected as part of the DRIVE and DRIVE-C study protocol through an eCRF using (c)2016 ennov clinical software version 7.5.720.1 (Paris, France).

Baseline characteristics were presented as frequencies and percentages for categorical variables, and medians and interquartile ranges (IQR) for continuous variables. Differences in baseline characteristics according to discordant results were tested using Fischer’s Exact test. We also used Bland–Altman plot analysis to highlight the differences between the assays, where the differences between the two assays are plotted against the results of the gold standard. Bland–Altman plot analysis was also realized after stratification by HCV genotype. All HCV viral loads in copies/mL were transformed into IU/mL by a factor of 0.4 [14]. Statistical analysis was performed using SAS Version 9.4 for Windows software (SAS Institute Inc., Cary, NC, USA).

## 3. Results

### 3.1. Study Population and Characteristics

The PWID population was almost exclusively male, with two-thirds of them having injected heroin for more than 10 years. They were also mainly poly-drug users, largely centered around methamphetamine consumption (Table 1). More than half were engaged in maintenance-assisted therapy. Out of the 600 samples, four repeatedly failed with the Xpert HCV VL assay with an error code “probe check failed”. These samples, hereafter referred to as CPD failure, all had a detectable HCV VL using the Light Power iVA HCV kit (mean: 4.79 ± 1.21 log_10_ IU/mL). The following analysis was performed on the 596 samples with valid Xpert HCV VL data.

### 3.2. Diagnostic Accuracy Analysis

Out of the 596 samples, 491 (82.4%) were positive with Xpert HCV VL with a median of 6.31 log_10_ IU/mL (95%CI: 5.41–6.75). Among them, 462 samples were also positive with Light Power iVA HCV with a median of 6.05 log_10_ IU/mL (95%CI: 5.12–6.57), providing a sensitivity of 94.1% (95%CI: 91.6–96.0), and giving 29 discordant VA HCV RNA results. Among the 105 negatives with Xpert HCV VL, 101 were also negative with Light Power iVA HCV, providing a specificity of 96.2% (95%CI: 90.5–98.95), and giving four CPD discordant HCV RNA results. The diagnostic accuracy was 94.5% (95%CI: 92.3–96.2) (Table 2 and Table 3).

As shown in Figure 1A, for most of the concordant results, Light Power iVA HCV quantification was lower than the Xpert HCV VL quantification. A total of 52 samples had more than one log difference (156 with a threshold of 0.5 log_10_ IU/mL). This difference was more noticeable for VL below 5 log_10_ IU/mL with 31% of the samples having more than one log difference (51% with a threshold of 0.5 log_10_ IU/mL). The AUC of the ROC curve of the assay dropped from 0.98 (95%CI: 0.97–0.99) for VL above 4 log_10_ IU/mL to 0.68 (95%CI: 0.58–0.79) for VL below 4 log_10_ IU/mL (Figure 1B,C). On the other hand, Xpert HCV VL was lower by more than one log to Light Power iVA HCV VL in 6 samples (12 with a threshold of 0.5 log_10_ IU/mL). These differences were independent of the HCV genotype (Figure 1A).

### 3.3. Factor Influencing the Discordant Results

Among the sociodemographic data, co-infections, or drug use variables, “being co-infected by HIV or HBV” were positively associated with VA discordant HCV RNA results when compared to the samples that were concordant between tests. Methadone use was associated with CPD failure/discordant HCV RNA results (Table 4).

## 4. Discussion

The accuracy of HCV RNA quantification is crucial for the diagnosis of HCV seropositive patients with chronic hepatitis C as well as for the assessment of the sustained virological response 12 weeks (SVR12) after the end of a DAA treatment. In this study, we showed that both the FDA- and CE-approved method, as well as a locally developed technique, were unsuccessful in accurately diagnosing 100% of cases with HCV-positive viremia.

The domestic assay had a good concordance rate with Xpert HCV VL, but sensitivity and specificity were below the WHO’s recommendations [4]. Furthermore, its lower limit of detection was at least one log too high to fit with the WHO’s recommendations [4]. As a consequence, its diagnostic accuracy rather fits with the purpose of chronic HCV infection diagnosis when at least 70% of the samples or more are positive. Indeed, the estimated predictive positive value (PPV) was 99%. Such HCV RNA prevalence is common among untreated HCV seropositive patients [15]. However, when the HCV RNA prevalence decreased down to 10% or 5% (Table 3), these estimated PPV dropped to 73% and 56%, respectively. Such HCV RNA prevalence is common when monitoring the efficacy of therapeutic treatment [16]. Furthermore, if the lower limit of detection for HCV diagnostic purposes is not a true issue during natural infection, the situation is far different at SVR12 when HCV viremia below 10^4^ IU/mL becomes more frequent [16].

The Xpert HCV VL assay was highly efficient in detecting HCV RNA among a set of HCV-infected samples, which were of genotype 6 for almost 40% of them. This result was recently reported by other researchers in a series of samples from Cambodia [17]. However, a total of eight samples were discordant with the domestic assay. We believe that these results are true false negative results because they were tested at least twice and on blood samples collected at different time-points. It was probably not a matter of threshold of detection as the mean HCV VL by Light Power iVA HCV assay was of 5.05 ± 1.6 log10 IU/mL. For four of them, the error message was “Probe check control” which encompasses reagent rehydration, probe integrity and dye stability. Would these blood samples contain inhibitors to which the Xpert technology was sensitive? We were not able to identify them among the drug used by the PWID. For the other four samples, the assays were completed but resulted as “not detectable”. Could the viral strains be too different from the consensus used in the Xpert HCV VL assay? Or could inhibitors also influence these steps? Unfortunately, we do not have evidence to discuss further. The false negative rate was estimated at above 1%, which may also have an impact on the efficacy of a delocalized testing campaign, especially among a volatile population such as PWID [18].

Much attention is currently being paid to the use of commercial NAT techniques with point-of-care access. Meta-analyses showed that these solutions perform as well as the techniques used in central laboratories with a sensitivity rate and diagnosis accuracy above 90% [19]. This is particularly true for the diagnosis of chronic hepatitis C, since the final aim is to adopt a test and treat strategy on the same day to reach elimination [20,21]. Nonetheless, this strategy will require significant resources and so will not rule out the use of “domestic” techniques in LMIC. It is very unlikely that LMIC will develop their own POC assay as it requests sophisticated technologies, which are frequently patented. Therefore, to prevent from inadequate use, “domestic” assays need to undergo a proper validation process [22] with a minimum number of steps for this to be achieved. US and European standards may be not affordable, but requirements listed in the MIQE guidelines (Minimum Information for publication of Quantitative real-time PCR Experiments, [23]) together with a comparison with gold standard assays on international sample panels could be easily implemented. A minimal validation plan including a regular independent monitoring of their performances (annually?) are definitively needed.

This study has some limitations. First, it was the result of a single site study. Although each step of the diagnostic followed good laboratory procedures, it would be interesting to replicate these results in another setting. Second, these conclusions were applied to PWID only and may not be true in other HCV-infected patients. Finally, the discordant Xpert negative HCV VL patients were not documented like the Xpert positive HCV VL ones, especially regarding HCV genotype.

## 5. Conclusions

Altogether, reaching HCV elimination by 2030 as targeted by the WHO’s recommendations [3] would require efficient HCV NAT at all steps of the HCV cascade of care. Domestic assays are indeed very useful for offering diagnostic treatment to a large number of people at an affordable cost. The domestic assay evaluated in the present manuscript had performance only compatible with the diagnostic of chronic hepatitis after a serologic pre-screening and not for therapeutic cure evaluation. The diagnostic performance of domestic assays clearly needs to be properly evaluated in order to use them effectively.

## Figures and Tables

**Figure 1 diagnostics-13-03456-f001:**
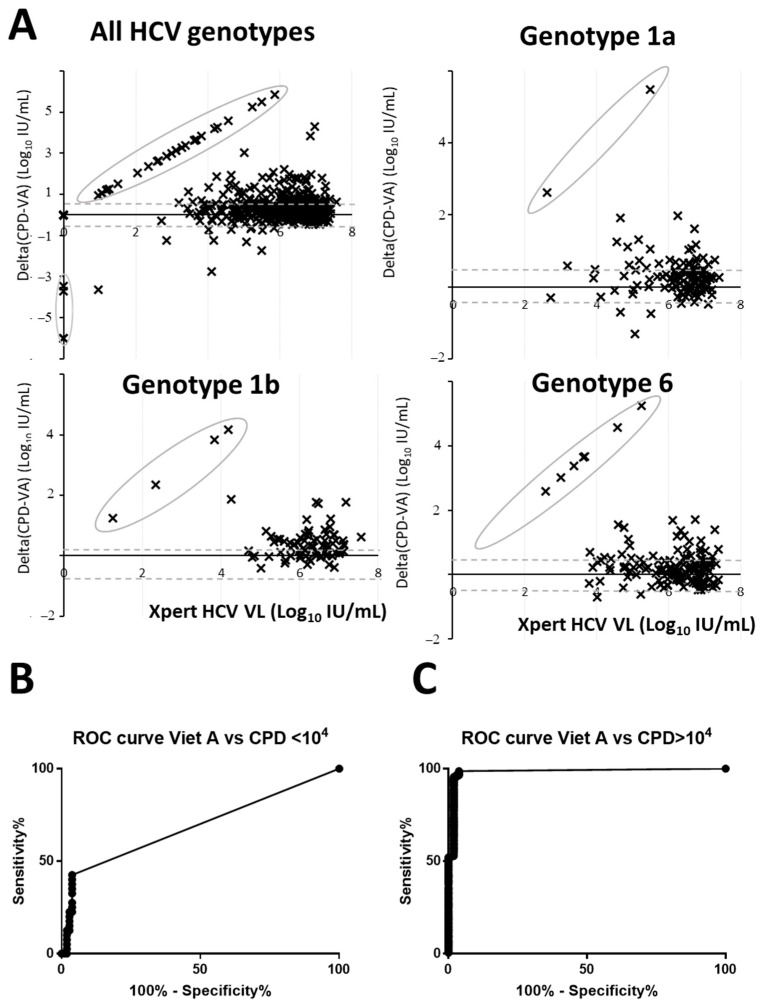
Bland–Altman plot and ROC analysis of HCV RNA quantification. (**A**) Quantification by Xpert HCV VL assay used as the gold standard are plotted on the *X*-axis while the log difference between Xpert HCV VL and Light Power iVA HCV was plotted on the *Y*-axis. Circled dots highlight either negative Light Power iVA HCV samples but Xpert HCV VL positive and vice versa. (**B**,**C**) ROC curves for VL below and above 4 log_10_ IU/mL. CPD stand for Xpert HCV VL assay and VA for Light Power iVA HCV assay.

**Table 1 diagnostics-13-03456-t001:** Characteristics of PWID with HCV positive serology.

	Patients with Positive HCV Serology (*n* = 600)
Socio-demographic data	
Gender, *n* (%) Male/Transgender	575 (95.8)
Age, mean [SD] Years	41.4 [7.9]
Co-infection, *n* (%)	
HIV	220 (36.7)
HBV	23 (5.9) *
Drug detected by urine test, *n* (%)	
Heroin	600 (100)
Methadone	383 (63.8)
Methamphetamine	175 (29.2)
Cannabis	8 (1.3)
Other drug consumption, self-report, *n* (%)	
Ketamine	23 (3.8)
Ecstasy	27 (4.5)
Amphetamines	13 (2.2)
Cocaine	2 (0.3)

HCV: hepatitis C virus; HIV: human immunodeficiency virus; HBV: hepatitis B virus; SD: standard deviation; * 210 missing data.

**Table 2 diagnostics-13-03456-t002:** Contingency table for the Light Power iVA HCV assay with Xpert HCV VL assay as reference.

		Xpert HCV VL		
		Positive	Negative	Total
Light Power iVA HCV VL	Positive	462	4	465
	Negative	29	101	131
	Total	491	105	596

HCV VL: hepatitis C virus viral load; iVA HCV VL: Light Power iVietA hepatitis C virus viral load.

**Table 3 diagnostics-13-03456-t003:** Evolution of the performance with different HCV prevalence (value (95%CI)).

HCV RNA Prevalence	82%	70%	10%	5%	1%
Positive predictive value	99.1% (97.7–99.7)	98.3% (95.7–99.3)	73.3% (51.2–87.8)	56.5% (33.2–77.3)	20.0% (8.7–39.5)
Negative predictive value	78.1% (71.5–83.6)	87.5% (83.0–90.9)	99.3% (99.0–99.5)	99.7% (99.5–99.8)	99.9% (99.9–99.9)
Accuracy	94.5% (92.3–96.2)	94.7% (92.6–96.4)	96.0% (94.1–97.4)	96.1% (94.2–97.5)	96.2% (94.3–97.6)

HCV: hepatitis C virus.

**Table 4 diagnostics-13-03456-t004:** Characteristics of the discordant results.

	CPD Failure (*n* = 4)	CPD Failure/Discordant (*n* = 8)	VA Discordant (*n* = 29)
Socio-demographic data			
Gender, *n* (%) male/transgender	4 (100)	8 (100)	27 (93.1)
Age, mean [SD] years	36 [9.17]	41.8 [8.55]	40.8 [8.36]
Co-infection, *n* (%)			
HIV	0 (0)	1 (12.5)	15 (51.7) *
HBV	°	£	5 (31.25) §*
HCV genotype	*n* = 0	*n* = 0	*n* = 14
GT1A			2 (14.3)
GT1B			4 (28.6)
GT2			1 (7.1)
GT3A/B			1(7.14)
GT6			6 (42.8)
Drug detected in urine, *n* (%)			
Heroin	4 (100)	8 (100)	29 (100)
Methadone	0 (0)	2 (25.0) *	19 (65.5)
Methamphetamine	1 (25.0)	2 (25.0)	5 (17.2)
Cannabis	1 (25.0)	1 (12.5)	0 (0)
Drug consumption, self-report, *n* (%)			
Ketamine	0 (0)	0 (0)	2 (6.8)
Ecstasy	0 (0)	0 (0)	2 (6.8)
Amphetamines	0 (0)	0 (0)	1 (3.4)
Cocaine	0 (0)	0 (0)	0 (0)

CPD: Xpert HCV VL assay; VA: Light Power iVA HCV assay; SD: standard deviation; GT: genotype. ° 4 missing data, £ 8 missing data, § 13 missing data, * *p* value < 0.05 using Fisher exact test comparing discordant versus non-discordant results.

## Data Availability

The data presented in this study are available upon request to the corresponding author.

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
