# Peer review of "HCV RNA Quantification by a Domestic Commercial Assay: A Case Study among People Who Inject Drugs in Vietnam"

_diagnostics, 2023, doi:10.3390/diagnostics13223456_

Round 1
Reviewer 1 Report
Comments and Suggestions for Authors
The article submitted for review raises the problem of HCV infection, which is very important for social health around the world. In the reviewer's opinion, the article is interesting, but the authors should improve a few points before the article is published:
- the consent number and the name of the bioethics committee should be added to the methodology
- the word sex in Tables 1 and 4 should be changed to gender
- figures, even in the form of additional materials, of the analysis results (amplification curve) of both tests would enrich the research
- the conclusions lack a sentence summarizing the potential use in diagnositics of the test analyzed in the research
Author Response
We thank the reviewer for helping us to improve the manuscript.
- the consent number and the name of the bioethics committee should be added to the methodology.
This information was present at line 311-315 under the section “Institutional Review Board Statement ». We duplicated this information in the M&M section as a 2.9 subsection.
- the word sex in Tables 1 and 4 should be changed to gender
The tables have been modified accordingly.
- figures, even in the form of additional materials, of the analysis results (amplification curve) of both tests would enrich the research
The amplification curves from the Genexpert are still available, while those from the VietA assay are not. Only datalog with Ct and standard curves were stored by the project local team. The computer associated with this instrument was replaced two or three years ago and we can't locate it.
- the conclusions lack a sentence summarizing the potential use in diagnositics of the test analyzed in the research
We added to the conclusion section the following: “The domestic assay evaluated in the present manuscript had performance only compatible with the diagnostic of chronic hepatitis after a serologic pre-screening and not for therapeutic cure evaluation. The diagnostic performance but their limitations of domestic assays in term of diagnos-tic performance clearly needs to be improved properly evaluated in order to use them effectively and in fine be able to reach the targets for HCV elimination.
Reviewer 2 Report
Comments and Suggestions for Authors
Dear Authors,
The manuscript entitled “HCV RNA quantification by a domestic commercial assay: a case study among people who inject drugs in Viet Nam”, addresses a relevant issue regarding the importance of adopting affordable NAT for detecting HCV infection and monitoring the efficacy of DAA therapy. My comments are listed below.
Page 1
Line 70 (also in line 80): Authors should specify what are the contaminants of the NAT they are referring to. It is also important to identify which step of NAT these contaminants interfere with (RNA extraction, RT, amplification...)?
Page 3 and throughout the text
Line 151: As two different methods are being compared, it is important that the units used to quantify HCV RNA are the same in both methods (RNA copies or IU, not a mixture of the two).
Page 5
Related to the previous comment, in line 194, the authors mentioned the median positive results for the Xpert HCV test, but I could not find the median positive results for the Light Power test.
Page 8
The authors state that “Meta-analyses showed that these solutions perform as well as, or even as well as, the techniques used in central laboratories”. This needs to be rephrased as it makes no sense.
Comments on the Quality of English LanguageMinor improvements
Author Response
We thank the reviewer for helping us to improve the manuscript.
Page 1
Line 70 (also in line 80): Authors should specify what are the contaminants of the NAT they are referring to. It is also important to identify which step of NAT these contaminants interfere with (RNA extraction, RT, amplification...)?
The following text was added:
- Line 70 “such as substances interacting directly with DNA or blocking the activity of the polymerase and reverse transcriptase or other PCR mixture components (e.g., MgCl2) (ref.)”
- Line 80 “(psychoactive drugs, polyphenols (tea), polysaccharides (algae, bivalves), antiretroviral drugs, … or any combination thereof) [11].”
Page 3 and throughout the text
Line 151: As two different methods are being compared, it is important that the units used to quantify HCV RNA are the same in both methods (RNA copies or IU, not a mixture of the two).
We totally agree with the reviewer. The text was modified with the use of the IU/mL unit throughout. The Y-axis of the figure #1 was wrongly labelled with cp/mL unit, this is now corrected.
Page 5
Related to the previous comment, in line 194, the authors mentioned the median positive results for the Xpert HCV test, but I could not find the median positive results for the Light Power test.
This value is now inserted in the text Line 199 as followed: “with a median of 6.05 log10 IU/mL (95%CI: 5.12-6.57)”
Page 8
The authors state that “Meta-analyses showed that these solutions perform as well as, or even as well as, the techniques used in central laboratories”. This needs to be rephrased as it makes no sens.
Thank you for pointing out this inconsistency. The text was modified as followed: “Meta-analyses showed that these solutions perform as well as, or even as well as, the techniques used in central laboratories with a sensitivity rate and diagnosis accuracy above 90%”
Reviewer 3 Report
Comments and Suggestions for Authors
This manuscript has compared the performances such as accuracy between the low-cost, easy-access HCV domestic assay and the widely accepted NAT test. The evaluation has concluded that even though the test threshold of domestic assay is high, its performance in terms of diagnosis purposes is comparable with the standard HCV RNA test, indicating the potential applicability of such domestic assay in less-developed areas of a country.
Overall, I believe this study is significant and urgent for promoting the diagnosis and treatment of HCV viral infection, and the manuscript is also well-organized with sufficient data and analysis. I do not have major concerns, but one minor point might be considered by the authors. Do you have any suggestions about how to optimize the current domestic assay(s) in order to increase their test accuracy? If so, please discuss and give your opinions in the manuscript.
Author Response
We thank the reviewer for helping us to improve the manuscript.
Thank you for these comments. We added in the discussion section the following text (line 279-288): “It is very unlikely that LMIC will develop their own POC assay as it requests sophisticated technologies, which are frequently patented. Therefore, to prevent from inadequate use, “domestic” assays need to undergo a proper validation process [22] with a minimum number of steps to be achieved. US and European standards may be not affordable but requirements listed in the MIQE guidelines (Minimum Information for publication of Quantitative real-time PCR Experiments, [23]) together with a comparison with a gold standard assays on international sample panels could be easily implemented. A minimal validation plan including a regular independent monitoring of their performances (annually?) are definitively needed.”